# Risk Factors of Postoperative Acute Pancreatitis and Its Impact on the Postoperative Course after Pancreaticoduodenectomy—10 Years of Single-Center Experience

**DOI:** 10.3390/life13122344

**Published:** 2023-12-15

**Authors:** Magdalena Gajda, Ewa Grudzińska, Paweł Szmigiel, Piotr Czopek, Cezary Rusinowski, Zbigniew Putowski, Sławomir Mrowiec

**Affiliations:** 1Department of Gastrointestinal Surgery, Medical University of Silesia, 40-752 Katowice, Poland; magdagajda20@poczta.fm (M.G.);; 2Center for Intensive and Perioperative Care, Jagiellonian University, 31-007 Cracow, Poland

**Keywords:** postoperative acute pancreatitis (PPAP), postoperative complications, postoperative pancreatic fistula (POPF)

## Abstract

Background: Clinically relevant acute postoperative pancreatitis (CR-PPAP) after pancreaticoduodenectomy (PD) is a complication that may lead to the development of local and systemic consequences. The study aimed to identify risk factors for CR-PPAP and assess the impact of CR-PPAP on the postoperative course after PD. Methods: The study retrospectively analyzed data from 428 consecutive patients who underwent PD at a single center between January 2013 and December 2022. The presence of increased amylase activity in plasma, above the upper limit of normal 48 h after surgery, was checked. CR-PPAP was diagnosed when accompanied by disturbing radiological features and/or symptoms requiring treatment. We investigated the relationship between the occurrence of CR-PPAP and the development of postoperative complications after PD, and possible predictors of CR-PPAP. Results: The postoperative follow-up period was 90 days. Of the 428 patients, 18.2% (n = 78) had CR-PPAP. It was associated with increased rates of CR-POPF, delayed gastric emptying, occurrence of intra-abdominal collections, postoperative hemorrhage, peritonitis, and septic shock. Patients who developed CR-PPAP were more often reoperated (37.17% vs. 6.9%, *p* < 0.0001)) and had increased postoperative mortality (14.1% vs. 5.74%, *p* < 0.0001). Soft pancreatic parenchyma, intraoperative blood loss, small diameter of the pancreatic duct, and diagnosis of adenocarcinoma papillae Vateri were independent risk factors for CR-PPAP and showed the best performance in predicting CR-PPAP. Conclusions: CR-PPAP is associated with an increased incidence of postoperative complications after PD, worse treatment outcomes, and an increased risk of reoperation and mortality. Pancreatic consistency, intraoperative blood loss, width of the duct of Wirsung, and histopathological diagnosis can be used to assess the risk of CR-PPAP. Amylase activity 48 h after surgery > 161 U/L is highly specific in the diagnosis of CR-PPAP.

## 1. Introduction

Pancreaticoduodenectomy (PD) is a technically demanding surgical procedure used for the treatment of lesions occurring in the head of the pancreas, ampulla of Vater, distal bile duct, and duodenum [1]. It is a surgery with a high risk of postoperative complications, reaching 50% according to some publications [2]. These complications include pancreatic fistula, delayed gastric emptying (DGE), and hemorrhage [2]. Postoperative acute pancreatitis (PPAP) is a phenomenon known to clinicians and has been the subject of numerous discussions among surgeons performing pancreatic surgery. It is suspected that it may influence the development of other postoperative complications, including the development of postoperative pancreatic fistula (POPF), which may lead to prolonged hospitalization, escalation of treatment costs, delayed use of adjuvant therapy, and may have a negative impact on overall survival [3]. Despite previous attempts to define PPAP, such as Connor’s research [4], only in 2022 did the International Study Group on Pancreatic Surgery (ISGPS) establish a commonly accepted definition of PPAP, as plasma amylase activity above the upper limit of normal persisting for at least 48 h after surgery [5]. It is particularly important to separate clinically insignificant hyperamylasemia (POH) and clinically significant PPAP (CR-PPAP) in the definition of PPAP. CR-PPAP is associated with the occurrence of disturbing radiological features in imaging studies and clinical symptoms requiring treatment [5]. To date, there are no recommendations for the early detection of PPAP, and there are no guidelines for its treatment. The aim of our study was to identify the risk factors for CR-PPAP and assess the impact of CR-PPAP on the postoperative course after PD procedures.

## 2. Materials and Methods

A single-center study was conducted on PD surgeries performed by one surgeon (SM) from January 2013 to December 2022 in the Department of Gastrointestinal Surgery, Medical University of Silesia in Katowice, Poland. The bioethical committee’s consent was waived due to the retrospective and anonymous character of the study.

After excluding 84 patients with incomplete clinical data (no amylase activity measurement 48 h after surgery), the data of all 428 patients operated on during the study period were retrospectively analyzed. Demographic data, details of surgical procedures (operative technique, palpation of pancreatic consistency), analysis of the postoperative course, amylase activity 48 h after surgery, and histological data of postoperative specimens were collected. The follow-up duration was 90 days.

PD was performed using the Whipple or the Traverso (pylorus-preserving) method. Pancreaticoenteric anastomosis was performed using the duct-to-mucosa method (n = 395, 92.3%) or intussusception (n = 33, 7.7%). In each patient, two peritoneal drains were placed (20 and 24 Fr) and removed, based on low amylase activity in the drain fluid on postoperative days (POD) 2–4. Perioperatively, a somatostatin analog (octreotide) was used selectively for pancreatic patients at a high risk of pancreatic fistula (soft consistency of the pancreatic parenchyma on palpation and/or narrow pancreatic duct < 2 mm).

In accordance with our institution’s scheme, blood samples were routinely obtained before and after surgery. Amylase levels in the serum were evaluated preoperatively and on postoperative days (POD) 1, 2, 3, and 5–7. Acute postoperative pancreatitis (PPAP) was defined according to the guidelines of the ISGPS as plasma amylase activity above the upper limit of normal persisting for at least 48 h after surgery [5]. PPAP was considered clinically relevant (CR-PPAP) when accompanied by disturbing radiological features (Table 1) and symptoms requiring treatment. PPAP was classified according to the ISGPS definition as type A (clinically insignificant POH) and CR-PPAP (clinically relevant types B and C). In the present study, the upper limit of normal for plasma pancreatic amylase activity measured 48 h after surgery was 98 U/L. Other PD-specific complications, including POPF [6], hemorrhage [7], DGE [8], and biliary anastomotic leak [9] were also defined according to ISGPS. All postoperative complications up to POD 90 were collected and categorized according to the Clavien–Dindo classification [10].

Currently we have no clear guidelines for CR-PPAP treatment. In the institution where the study was conducted, patients diagnosed with CR-PPAP were treated with a somatostatin analog (octreotide) at a dose of 100 µg 3 × 1 mL SC for 3–7 days, parenteral nutrition (i.v., for 7–10 days), and, in the event of an increase in C-reactive protein (CRP) level above 200 mg/L, preventive antibiotic therapy (piperacillin + tazobactam at a dose of 3 × 4.5 g i.v., for 10 days) was administered.

A Computed Tomography (CT) scan of the abdominal cavity and pelvis with orally and intravenously administered contrast was performed in patients with elevated laboratory inflammatory parameters: CRP > 200 mg/L, Procalcitonin (PCT) > 2.5 ng/mL, and/or in patients with disturbing physical symptoms (e.g., abdominal pain, fever) to expand diagnostics on POD 3–12 (n = 273, 63.8%).

Statistical analysis was performed using R software version 4.3.1 (Beagle Scouts). Continuous variables are reported as median and interquartile range (i.q.r.), and categorical variables as frequencies and proportions (percent). Differences between continuous variables were assessed using the Mann–Whitney U test; for categorical variables, differences were assessed using the χ^2^ test. Receiver operating characteristic (ROC) curves were used to analyze the association of risk factors and the occurrence of CR-PPAP. Multivariate logistic regression was performed to assess potential risk factors for CR-PPAP. The analyzed factors included: consistency of the pancreatic parenchyma, width of the pancreatic duct, histopathological diagnosis of ampulla of Vater cancer, duration and type of the procedure, intraoperative blood loss, use of neoadjuvant chemotherapy, gender, and BMI (body mass index). The factors for multivariate analysis were selected based on expert knowledge about the risk factors for the development of CR-PPAP according to ISGPS [5] and other recognized publications [4,6].

## 3. Results

### 3.1. Patient Demographics

A total of 512 patients underwent PD during the study period. After excluding patients with missing data on plasma amylase (n = 84), a total of 428 (83.6%) patients were included in the study. Demographic data are presented in Table 2.

### 3.2. Postoperative Complications

The postoperative course during 90 PODs was analyzed. Postoperative complications occurred in 77 (98.71%) patients with CR-PPAP compared to 36 (41.37%) patients without CR-PPAP (*p* < 0.0001).

The relationship between the occurrence of CR-PPAP and the development of postoperative complications after pancreaticoduodenectomy was examined (Table 3).

Out of 428 patients, PPAP occurred in 38.5% (n = 165), of whom 18.2% (n = 78) had CR-PPAP. Patients who developed CR-PPAP had an increased incidence of severe complications according to the Clavien–Dindo scale (*p* < 0.0001). CR-PPAP was associated with an increased rate of CR-POPF (*p* < 0.0001), DGE (*p* < 0.0001), a higher occurrence of abdominal collections (*p* < 0.0001) and abscesses (*p* < 0.0001), as well as postoperative hemorrhage (*p* < 0.0001), peritonitis (*p* < 0.0001), and septic shock (*p* < 0.0001) (Figure 1).

A postoperative CT scan was performed in 273 (63.8%) patients after PD. CR-PPAP was significantly associated with diffuse or localized inflammatory enlargement of the pancreatic stump, inflammatory changes in the peripancreatic fat tissue, intra- and/or peripancreatic fluid collections, necrosis of the pancreatic parenchyma, peripancreatic necrosis, leakage of pancreatic anastomosis, peritonitis, and bleeding (Table 4).

A total of 47 patients after PD procedures underwent reoperation, of which 29 were patients with CR-PPAP (Table 5). All reoperated patients (both with and without CR-PPAP) required abdominal lavage and drainage, as well as evacuation of fluid collections/abscesses. Patients who developed CR-PPAP were more likely to require reoperation (n= 29, 37.17% vs. n= 6, 6.9%, *p* < 0.0001), and had increased postoperative mortality (n = 11, 14.1% vs. n = 5, 5.74%, *p* < 0.0001). Patients who developed CR-PPAP had a prolonged hospital stay compared to patients without CR-PPAP (24 vs. 15 days, *p* < 0.0001).

### 3.3. Amylase Values and the Development of CR-PPAP

Amylase values at 48 h after PD were analyzed using ROC analysis to find the threshold value for the development of CR-PPAP. An amylase value > 161 U/L has a sensitivity of 98.7% and a specificity of 83.7% for the diagnosis of CR-PPAP (Figure 2 and Figure 3). An amylase value > 320 U/L indicates a 50% probability of CR-PPAP.

### 3.4. Risk Factors for the Development of CR-PPAP

Selected clinical features were analyzed to identify factors influencing the development of CR-PPAP. The multivariate analysis examining the risk factors for CR-PPAP is presented in Table 6.

Factors such as soft consistency of the pancreatic parenchyma (OR 4.32; 95% CI 1.51–12.3; *p* = 0.0061), intraoperative blood loss (OR 0.99; 95% CI 0.99–0.99; *p* = 0.0349), Wirsung’s duct diameter (OR 0.47; 95% CI 0.35–0.64; *p* < 0.0001), and diagnosis of adenocarcinoma of the ampulla of Vater (OR 3.17; 95% CI 1.69–5.94; *p* = 0.0003) were independent risk factors for CR-PPAP and showed the best performance in predicting CR-PPAP (AUC = 0.945, area under the ROC curve 0.782) (Figure 4).

## 4. Discussion

Postoperative acute pancreatitis (PPAP) in recent years has become a topic of great interest among pancreatic surgeons. The lack of a uniform definition for PPAP until 2022 has adversely affected the study of this phenomenon, making comparisons of results between studies difficult. In our work, we used the ISGPS definition of PPAP, which considers only one laboratory parameter—elevated postoperative serum amylase levels lasting for at least 48 h [5]. In studies performed before 2022, according to some authors, other parameters such as CRP [4,11,12,13], lipase [14], or urinary trypsinogen-2 (U-TRP-2) [4,15] best reflected the development and course of PPAP. Some researchers [16] used the classification developed by Connor [4] in 2016, which was based on increased serum amylase or lipase activity above the upper limit of normal on POD 0–1, increased serum levels CRP > 180 mg/L on POD 2, and clinical course. In turn, many authors [17,18] relied on the Atlanta classification of acute pancreatitis, commonly known to clinicians [19], the diagnosis of which is based on the identification of at least two of the following symptoms: pain in the upper abdomen; amylase and/or lipase activity higher than three times the institutional norm; and symptoms of inflammation in the pancreas and/or pancreatic area based on additional imaging tests. Because of this discrepancy in PPAP definitions, it is difficult to compare results concerning the occurrence of this complication. Murakawa et al. reported a 58.9% occurrence of postoperative pancreatitis after PD, applying Connor’s definition of hyperamylasemia on POD0 and1 [20]. In our results, PPAP occurred in 38.5% of patients, with 18.2% of CR-PPAP. This is more in line with Bonsdorff’s results [21], involving 508 patients undergoing PD, where PPAP was noted in 39.8% and CR-PPAP in 17.9%. PPAP was defined as an elevated serum amylase level on POD 1, considered clinically relevant when accompanied by a CRP elevation to 180 mg/L or more on POD 2.

Despite the various definitions of PPAP, the development of this phenomenon is generally considered a significant threat in the postoperative course after PD procedures [12,16]. Our study also shows that the occurrence of CR-PPAP significantly increases the risk of postoperative complications, which translates to increased postoperative mortality (14.1% vs. 5.74%, *p* < 0.0001). In the available literature, the presence of CR-PPAP, regardless of the adopted definition, has always been associated with a higher risk of other postoperative complications [16,17,18,22]. The close relationship between CR-PPAP and the development of CR-POPF is particularly interesting [17,23,24]. In both complications, hyperamylasemia occurs in the early postoperative days. Moreover, the risk factors for CR-PPAP (pancreatic parenchyma consistency, intraoperative blood loss, width of the duct of Wirsung, and histopathological diagnosis of ampulla of Vater carcinoma) overlap with the risk factors for CR-POPF after PD according to the Fistula Risk Score (FRS) [25]. In our study, out of 78 patients with CR-PPAP, 77 were also diagnosed with CR-POPF. These results support the hypothesis of a possible physio-pathological connection between PPAP and POPF, although both complications may occur independently [26,27]. There is no clear explanation in the literature for the cause-and-effect mechanism between CR-PPAP and CR-POPF. It is suspected that one of the most important factors is postoperative pancreatic ischemia. According to the hypothesis, anastomosis leakage develops as a consequence of necrosis in the pancreatic stump [11,28]. Proper surgical technique, including the use of fine sutures and delicate knotting for the pancreatic anastomosis, is also considered important [29].

CR-PPAP in our study is associated with a significantly prolonged hospital stay (24 vs. 15 days, *p* < 0.0001). These data are consistent with the available literature [3]. Identification of CR-PAAP risk factors and initiating early treatment could reduce the occurrence of this complication, shortening the hospitalization duration. The present study did not assess treatment costs, but shortening hospitalization time thanks to early diagnosis of patients at risk of CR-POPF could contribute to cost reduction. In turn, the lack of CR-PPAP risk factors in patients may help identify a group of patients with a low risk of postoperative complications, who would potentially qualify for the implementation of a fast recovery path, with earlier initiation of feeding and faster removal of abdominal drains.

According to Chen et al., who performed a large retrospective study on 1465 patients, female gender is a risk factor for CR-PPAP. It is important to note that in their study, PPAP was defined as amylase serum levels above normal on POD 1 [30]. In our study, female gender did not show a significant impact on the development of this complication. This may be because of differences in sample size and/or differences in PPAP definition. Similarly, in another study, postoperative hyperamylasemia (measured on POD 0) had a positive correlation with patients’ BMI [17]. In our study, BMI did not show any significant correlation with PPAP.

The assessment of the consistency of pancreatic parenchyma as a risk factor for PPAP, based on the operator’s subjective assessment, may be controversial. A more accurate and objective test could be preoperative pancreatic elastography [31], although it is not widely available. Another proposal to objectify the assessment of pancreatic parenchyma involves intraoperative histopathological examination of the density of acinar cells in the margins of the resected pancreas [32]. Some authors suggest that a high-risk pancreas with a high density of acinar cells is susceptible to both immediate leakage of pancreatic fluid rich in proteases and the development of pancreatitis in the remaining gland as a result of ischemia and/or mechanical manipulation [33]. These data highlight the importance of appropriate pancreatic texture assessment for the diagnosis and close monitoring of patients at higher risk of developing PPAP after PD. However, this also involves additional costs and could pose logistical challenges. This issue certainly requires further research.

The retrospective nature of this study is definitely a weakness, and prospective studies on larger patient groups are needed to further examine the risk factors and effects of PPAP. Another weak point may be single-center nature of this study [34]. Its results may not be generalized to other centers with different volumes of patients and variously experienced surgeons. On the other hand, the fact that all operations are carried out following one standard and by one operator, removes potential differences in results that could arise from various experience levels and operational techniques. At the same time, slight modifications in the surgical technique over the 10-year period (two types of pancreatic anastomoses) could not be avoided. According to some publications, the technique of pancreatico-enteric anastomosis (mucosa-duct vs. intussusception) may influence the occurrence of CR-POPF [35].

The issue of PPAP prevention and treatment remains unresolved. Long-established intensive treatment for acute pancreatitis, including antibiotic therapy, intensive fluid therapy in the first days after diagnosis, and parenteral or enteral nutrition, is usually implemented. Therapeutic success strongly depends on early diagnosis and treatment administration. However, in a postoperative setting, common pancreatitis symptoms (such as abdominal pain, nausea, and elevated temperature) may be erroneously attributed to other postoperative issues. The volume of administered intravenous fluids remains controversial, as it may affect the healing of anastomoses, which is an issue absent in non-postoperative settings [36,37].

In our study, octreotide was used perioperatively in selected cases where the soft consistency of the pancreatic parenchyma found intraoperatively and/or a narrow pancreatic duct (<2 mm) was considered to elevate the risk of postoperative PPAP and POPF [38]. The subjective choice of octreotide-treated patients in our study could certainly influence the results and introduce bias. Octreotide is a somatostatin analogue that inhibits exocrine secretion of the pancreatic remnant. It is not recommended for routine use in patients undergoing PD because of conflicting study results. Some results indicate its efficacy in reducing the incidence of postoperative complications such as POPF, intraabdominal fluid collections, and PPAP, without any difference in mortality rate [38,39,40]. Others show no benefit or even a negative impact on PD outcomes [41]. There are other reports showing that prophylactic administration of the drug ulinastatin reduces the levels of amylase in serum and intraabdominal drain fluid, and reduces the incidence of postoperative pancreatitis after pancreaticoduodenectomy though this drug is currently not widely used [42]. Therefore, no recommendations are available for pharmacological prevention of PPAP. Another method to avoid potentially fatal PPAP and POPF is upfront total pancreatectomy. It is accepted in selected cases with a higher risk of pancreatic anastomosis leakage due to soft pancreatic tissue and a narrow MPD. However, in the absence of other indications (e.g., disease affecting the entire gland), there are no uniform recommendations, and the decision is made by the surgeon [43]. In our department, upfront total pancreatectomy is occasionally performed in selected cases.

In advanced cases with extensive necrosis and concomitant sepsis, the only treatment method seems to be a reoperation [27,44,45]. In our study, 37.17% of patients with CR-PPAP underwent reoperation compared with 6.9% of patients without CR-PPAP. The most frequently performed procedures in reoperated patients with CR-PPAP included evacuation of reservoirs and abscesses from the peritoneal cavity (37.17%), drainage of the peritoneal cavity (37.17%), and distal resection of the pancreatic stump with splenectomy (25.64%). We rarely decided to perform pancreatic anastomosis again (5.12%). Our treatment course is similar to procedures described in the available literature. In the Rudis study, the development of PPAP combined with POPF type C almost always led to the patient’s death. The author emphasizes that separation of pancreatic anastomosis and drainage procedures is usually insufficient [27]. An appropriate but risky option for early revision in a patient with suspected PPAP is a rescue total pancreas removal with splenectomy [27]. In late revisions, the surgical field is significantly altered by PPAP, and the mortality rate after this type of procedure is very high. Total pancreatectomy may be of significant benefit when performed as soon as possible after the diagnosis of potentially fatal PPAP [45]. In each case, the decision to perform total pancreatectomy is very difficult and depends on the experience of the surgeon [27]. In a recent study already based on the new ISGPS definition of PPAP, the risk of a rescue complete pancreatectomy in patients after PD was retrospectively assessed. Postoperative hyperamylasemia on POD 1 (regardless of the presence of CR-PPAP) was identified as an independent risk factor of complete pancreatectomy [29]. A reoperation with total pancreatectomy has a high mortality rate but serves as an emergency procedure to deal with potentially fatal complications of PD [44,45].

## 5. Conclusions

CR-PPAP is a potentially life-threatening complication that is associated with an increased incidence of postoperative complications after pancreaticoduodenectomy, which worsens treatment outcomes, and increases the risk of reoperation and postoperative mortality. It is an independent risk factor for the development of CR-POPF, DGE, intraperitoneal collections and abscesses, hemorrhage, peritonitis, and septic shock after PD procedures.

An amylase value > 161 U/L 48 h after pancreaticoduodenectomy has high sensitivity (98.7%) and specificity (83.7%) in predicting CR-PPAP.

Factors such as the consistency of the pancreatic parenchyma, intraoperative blood loss, diameter of the Wirsung duct, and histopathological diagnosis can be used to assess the risk of CR-PPAP. The risk factors for CR-PPAP overlap with the risk factors for CR-POPF, so the occurrence of CR-PPAP may relate to the occurrence of CR-POPF.

Further research is needed, focusing on the currently missing guidelines on how to prevent and treat this complication.

## Figures and Tables

**Figure 1 life-13-02344-f001:**
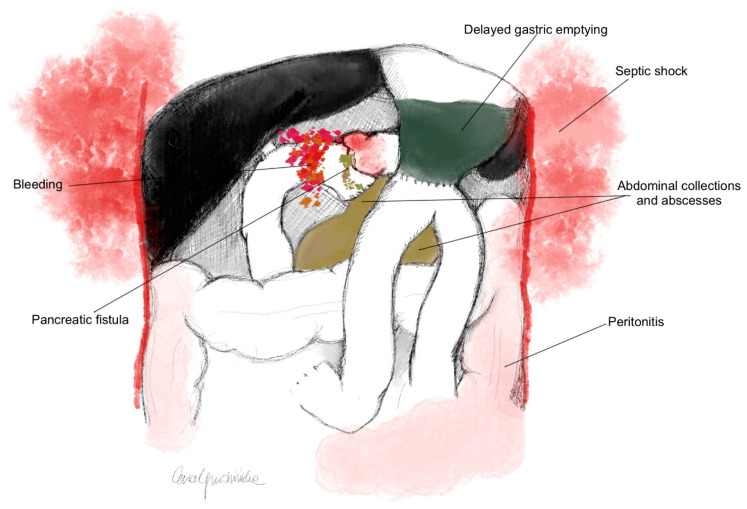
Postoperative complications after PD associated with the development of CR-PPAP (*p* < 0.0001).

**Figure 2 life-13-02344-f002:**
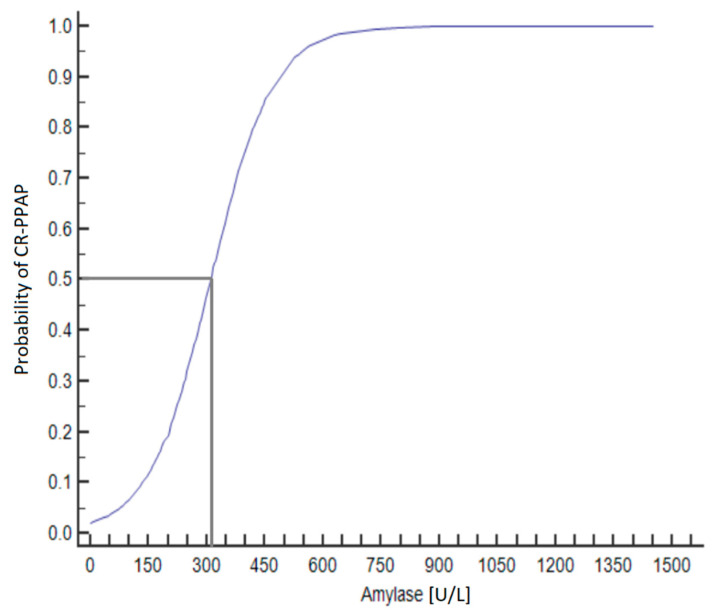
Predicted probability of CR-PPAP based on amylase values at 48 h after PD. 50% risk of developing CR-PPAP with amylase activity > 320 U/L.

**Figure 3 life-13-02344-f003:**
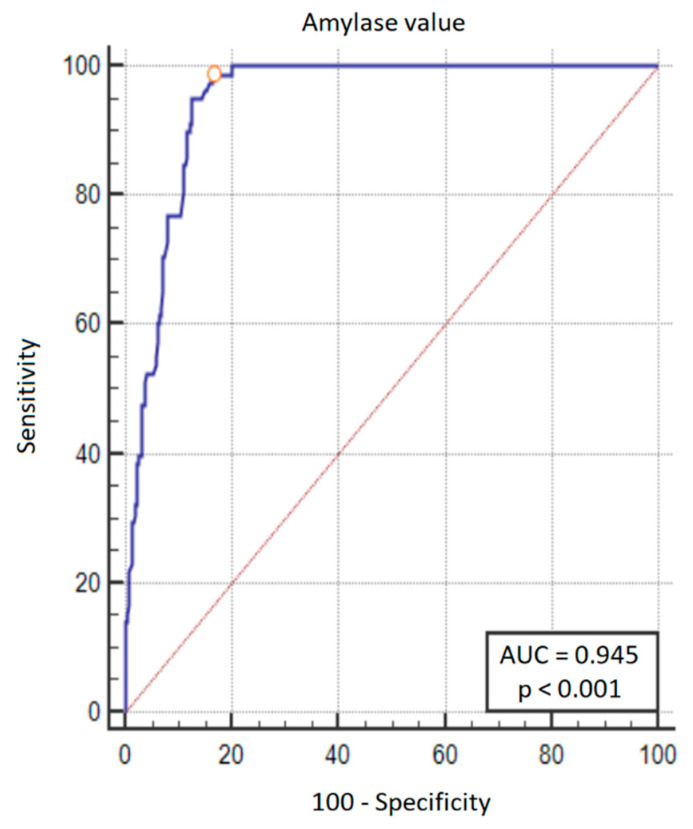
ROC curve plot for predicting the development of CR-PPAP. In addition to the ROC curve (blue line), the diagonal (red line) is also marked on the graph—this is the theoretical line of classification made by the random model. Orange circle on the graph is the threshold of amylase activity that showed the best trade-off between sensitivity and specificity—this value was > 161 U/L (sensitivity 98.7%, specificity 83.7%).

**Figure 4 life-13-02344-f004:**
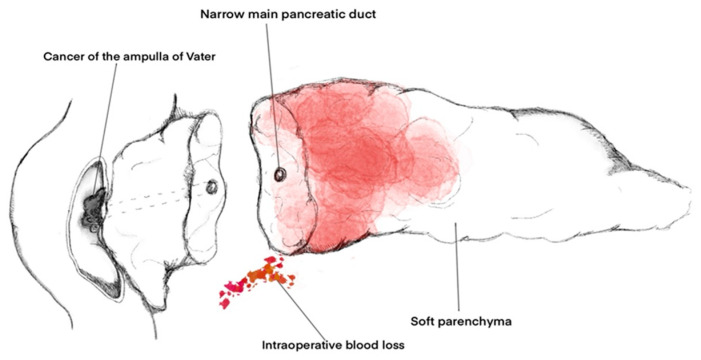
Independent factors for the development of CR-PPAP after PD.

**Table 1 life-13-02344-t001:** Characteristics of PPAP in the study population (*—upper limit of normal for the study institution).

**Pancreaticoduodenectomy (PD)** **n = 428 (100%)**	
**Postoperative Acute Pancreatitis (PPAP)** **Increase in Serum Amylase Activity > 98 U/L *** **n = 165 (38.55%)**	
**Clinically Relevant PPAP (CR-PPAP)** **n = 78 (18.22%)** associated disturbing radiological features: diffuse or localized inflammatory enlargement of the pancreatic stump (interstitial parenchymal edema), inflammatory changes in the peripancreatic fat tissue, intra- and/or peripancreatic fluid collections, necrosis of the pancreatic parenchyma and/or peripancreatic necrosis.	
**Type B** **n = 51 (11.91%)**	**Type C** **n = 27 (6.3%)**
Mild or moderate complications that may require emergency medical or minimally invasive treatment.	Severe life-threatening complications that require surgical intervention and may lead to organ failure and/or death.

**Table 2 life-13-02344-t002:** Patient demographics, perioperative data, and histopathological diagnosis of 428 patients undergoing pancreaticoduodenectomy (NET—pancreatic neuroendocrine tumor, IPMN—intraductal papillary mucinous neoplasm).

Demographics	n or Median (i.q.r.)
Age (years)	66 (60–70)
Sex (male:female)	189:239
BMI (kg/m^2^)	24.4 (21.9–26.4)
Type of operation	
Whipple procedure	235 (54.9)
Traverso procedure	193 (45.1)
Pancreas consistency	
Soft pancreas consistency	175 (40.9)
Normal pancreas consistency	222 (51.7)
Hard pancreas consistency	36 (8.4)
Pancreatic duct	
Pancreatic duct diameter (mm)	2 (2–3)
Typical location of pancreatic duct	344 (80.4)
Posterior location of pancreatic duct	80 (18.7)
Non-visible location of pancreatic duct	4 (0.9)
Other information	
Intraoperative blood loss (mL)	300 (300–400)
Procedure duration (min)	445 (420–490)
Neoadjuvant therapy (n)	145 (33.9)
Histopathological diagnosis	
Ductal pancreatic adenocarcinoma	263 (61.4)
High-grade adenoma Vateri	4 (0.9)
Adenocarcinoma Vateri	81 (18.8)
Duodenal Adenocarcinoma	5 (1.1)
Biliary Adenocarcinoma	8 (1.9)
Chronic pancreatitis	9 (2.1)
NETs	13 (3.0)
IPMN	15 (3.5)
Other diagnoses	30 (7.0)

**Table 3 life-13-02344-t003:** Association between the development of CR-PPAP and postoperative complications. (Statistical significance: *p* < 0.05).

Type of Complication	Patients without CR-PPAP n = 87 (100%)	Patients with CR-PPAP n = 78 (100%)	*p*
CR-POPF	8 (9.19%)	77 (98.71%)	<0.0001
Abdominal collection	36 (41.37%)	77 (98.71%)	<0.0001
Abscess	5 (5.74%)	37 (47.43%)	<0.0001
Peritonitis	7 (8.04%)	21 (26.92%)	<0.0001
Septic Shock	3 (3.44%)	19 (24.35%)	<0.0001
Bleeding	4 (4.59%)	14 (17.94%)	<0.0001
DGE	15 (17.24%)	56 (71.79%)	<0.0001
Biliary leakage	21 (24.13%)	9 (11.53%)	0.0835
Enteral leakage	1 (1.15%)	0 (0%)	0.2571
Peripheral thrombosis	6 (6.9%)	8 (10.25%)	0.0752
Intestinal obstruction	11 (12.64%)	12 (15.38%)	0.4564
Dehiscence of the postoperative wound	9 (10.34%)	5 (6.41%)	0.2782

**Table 4 life-13-02344-t004:** Radiological features in CT examination after PD (*—multiple features could be present in patients). (Statistical significance: *p* < 0.05).

Radiological Features in CT Scan *	Patients without CR-PPAP N = 87 (100%)	Patients with CR-PPAP N = 78 (100%)	*p*
Diffuse or localized inflammatory enlargement of the pancreatic stump	10 (11.49%)	78 (100%)	<0.0001
Inflammatory changes in the peripancreatic fat tissue	8 (9.19%)	78 (100%)	<0.0001
Intra- and/or peripancreatic fluid collections	36 (41.37%)	77 (98.71%)	<0.0001
Necrosis of the pancreatic parenchyma	1 (1.14%)	27 (34.61)	<0.0001
Peripancreatic necrosis.	2 (2.29%)	22 (28.2%)	<0.0001
Leakage of pancreatic anastomosis	1 (1.14%)	27 (34.61)	<0.0001
Leakage of the biliary anastomosis	21 (24.13%)	9 (11.53%)	0.0835
Leakage of the enteral leakage	1 (1.15%)	0 (0%)	0.2571
Peritonitis	7 (8.04%)	21 (26.92%)	<0.0001
Bleeding	4 (4.59%)	14 (17.94%)	<0.0001

**Table 5 life-13-02344-t005:** Reoperation after PD (*—treatment could involve more than one procedure). (Statistical significance: *p* <0.05).

Type of Procedure *	Patients without CR-PPAP n = 87 (100%)	Patients with CR-PPAP n = 78 (100%)	*p*
Abdominal lavage and drainage	6 (6.9%)	29 (37.17%)	<0.0001
Evacuation of fluid collections/abscesses	6 (6.9%)	29 (37.17%)	<0.0001
Re-anastomosis	4 (4.59%)	4 (5.12%)	0.4561
Wirsungostomy	1 (1.15%)	5 (6.41%)	<0.0001
Distal pancreatectomy with splenectomy	2 (2.3%)	20 (25.64%)	<0.0001
Bleeding management	4 (4.59%)	14 (17.94%)	<0.0001

**Table 6 life-13-02344-t006:** Risk factors for the development of CR-PPAP after PD. (Statistically significant factors were marked).

	*p*	Odds Ratio (OR)	95% CI
Procedure duration	0.3036	1.1982	0.8490–1.6910
**Soft pancreas consistency**	**0.0061**	**4.3239**	**1.5189–12.3086**
Neoadjuvant therapy	0.1511	0.6082	0.3084–1.1992
Type of operation	0.4731	1.2247	0.7040–2.1305
BMI	0.0517	1.4336	0.9973–2.0609
**Blood loss**	**0.0349**	**0.9977**	**0.9955–0.9998**
**Pancreatic duct diameter**	**<0.0001**	**0.4795**	**0.3555–0.6468**
**Adenocarcinoma papille Vateri**	**0.0003**	**3.1750**	**1.6949–5.9476**
Female sex	0.1571	0.6665	0.3799–1.1693

## Data Availability

Data are available from the corresponding author upon reasonable request.

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
