# Peer review of "Risk Factors of Postoperative Acute Pancreatitis and Its Impact on the Postoperative Course after Pancreaticoduodenectomy—10 Years of Single-Center Experience"

_life, 2023, doi:10.3390/life13122344_

Round 1

Reviewer 1 Report

Comments and Suggestions for Authors

thank you for allowing me to review this monocentric retrospective study. the manuscript is well written and illustrated. the main objective is clearly stated and the inclusion criteria. the definition of postoperative acute pancreatitis is clearly stated and referenced. authors excluded more than 15% of the workforce due to lack of data. the manuscript would gain legitimacy if these patients were included and statistical analysis uses multiple imputation with missing strings taking into account these missing data. 

the authors report a significant increase in morbidity including severity in patients with post-operative pancreatitis. the authors also identified independent risk factors. it would be desirable to have a table of both uni- and multivariate statistical analysis to know which factors were included in the multivariate analysis. 

However, the authors do not discuss the impact of their risk factors. we would have liked to know if these results could influence their future management; namely, no pancreatic anastomosis to avoid fistula following pancreatitis; preventive treatment at the intervention in case of risk factors ...  

Author Response

Dear Reviewer, thank you for the thorough review of our work. Below you will find our answers to the comments, we did our best to improve the manuscript according to your suggestions:

thank you for allowing me to review this monocentric retrospective study. the manuscript is well written and illustrated. the main objective is clearly stated and the inclusion criteria. the definition of postoperative acute pancreatitis is clearly stated and referenced. authors excluded more than 15% of the workforce due to lack of data. the manuscript would gain legitimacy if these patients were included and statistical analysis uses multiple imputation with missing strings taking into account these missing data.

- 84 patients were excluded from our study who did not have their amylase values determined postoperatively. Since the definition of CR-PPAP used in our study is based on the amylase value 48 hours after surgery, this group of patients could not be included in our analysis.

We have added an additional explanation (lines 62-63).

The authors report a significant increase in morbidity including severity in patients with post-operative pancreatitis. the authors also identified independent risk factors. it would be desirable to have a table of both uni- and multivariate statistical analysis to know which factors were included in the multivariate analysis.

- Thank you for this comment. Our selection of factors for multivariate analysis is based on expert knowledge about the risk factors for the development of CR-PPAP according to ISGPS [literature reference: 5] and other recognized publications [literature references: 4,6,17,25,35], and not on p-value screening. We analyzed factors such as the soft consistency of the pancreatic parenchyma, small (<2 mm) width of the pancreatic duct, histopathological diagnosis of the ampulla of Vater cancer, duration and type of the procedure, intraoperative blood loss, use of neoadjuvant chemotherapy, gender and BMI (body mass index). Selecting of the risk factors (predefined method) was based on this article: https://pubmed.ncbi.nlm.nih.gov/27896874/

We now added an appropriate explanation in our study – lines 109-114.

However, the authors do not discuss the impact of their risk factors. we would have liked to know if these results could influence their future management; namely, no pancreatic anastomosis to avoid fistula following pancreatitis; preventive treatment at the intervention in case of risk factors…

- Currently, there are no generally recognized guidelines on how to prevent CR-PPAP. In our study, octreotide was used perioperatively in selected cases, where the PPAP and POPF risk was considered high due to the pancreatic parenchyma intraoperatively assessed as soft and/or narrow pancreatic duct (<2mm) was found. However, there are conflicting results of studies, with some showing its efficacy in reducing the incidence of postoperative complications, such as POPF, intraabdominal fluid collections, and PPAP, without any difference in mortality rate and others postulating its negative impact on PD results.

Perhaps intensified postoperative supervision and prophylactic use of octreotide in selected patients with risk factors for the development of CR-PPAP mentioned in our study could contribute to the reduction of this complication. We shortly discuss this suggestion.

Upfront total pancreatectomy to completely avoid the pancreatic anastomosis creation and PPAP risk is accepted in selected cases. In patients with no other indications for total pancreatectomy present apart from the PPAP/ POPF risk factors (soft tissue, narrow MPD), the decision is made intraoperatively by the surgeon without any general recommendations. In our department upfront total pancreatectomy is performed at the discretion of the operating surgeon in selected high-risk cases. We added this to our discussion.

Reviewer 2 Report

Comments and Suggestions for Authors

The present study addresses one of the most feared complications after pancreaticoduodenectomy: postoperative acute pancreatitis. For a long time, there was no clear definition for acute postoperative pancreatitis after pancreaticoduodenectomy, being included in the broad spectrum of postoperative pancreatic fistula. However, the International Study Group for Pancreatic Surgery recently proposed a clear definition and stratification for such complications. Data about predictors of postoperative acute pancreatitis after pancreaticoduodenectomy and the potential clinical implications of such complications are scarce. Although it is a single-center study, the cohort includes a relatively high number of patients for analysis. So, the present study would be interested in the journal's readers. The study is well-designed, and it has scientific sound, but a few improvements of methods and other corrections should be made to see if the methods are correctly used and if the results support the conclusions:

Please consider putting Table 1 in a more friendly design to understand the provided data better. Maybe providing only the types of CR-PPAP is enough in Table 1.

Please consider defining in the text at first use CT, CRP, POD, and PCT.

How was assessed the pancreas consistency as soft, normal, and hard? What was the difference between soft and normal?

Please consider providing data in Table 3 as both numbers (percentages) for better understanding. What does it mean "fluid reservoir"? Why was it regarded as statistical significance at p values <0.0001, as suggested in Table 3 headline? In Table 3, DGE, bleeding was CR or overall?

In lines 122-124, it is said that patients with CR-PPAP have had an increased incidence of severe complications compared with patients with no CR-PPAP. Please consider providing percentages for each group.

In line 127, I assume the reference is to Table 3, not Table 4. Please check.

Table 4 data should be discussed and explained in the text, and p values should be provided.

The data provided in Table 5 are weird. For example, What does "reoperation 61.7% vs. 38.29%" mean? The same issue applies to 90-day mortality and types of procedures. Please check and put the correct values.

Regarding the indications for re-exploration, was it only for collections no bleeding? (as one might assume from lines 133 – 135). Please clarify.

On what basis were selected the factors to be explored as potential predictors for the development of CR-PPA? A few potential important factors such as ASA score, cardiovascular co-morbidities, diabetes, jaundice, cholangitis, preoperative biliary drainage,  neuroendocrine vs. adenocarcinoma pathology, type of distal pancreatic stump anastomosis, CRP, PCT, etc, appear to be overmissed. Please explain.

It is uncommon that all the statistically significant factors from the univariate analyses were found to be independent predictors. Please comment.

Figure 4 appears not to illustrate the ampulla of Vater cancer location correctly.

Based on the results of the present study, what is the suggested value for clinical decision-making?

The references should be provided in the format the journal requires.

Author Response

Dear Reviewer, thank you for the insightful and detailed comments. We did our best to correct the existing errors and to improve the quality of our work based on your review:

Please consider putting Table 1 in a more friendly design to understand the provided data better. Maybe providing only the types of CR-PPAP is enough in Table 1.

- Table 1 has been corrected according to the recommendations.

Please consider defining in the text at first use CT, CRP, POD, and PCT.

- All abbreviations are now defined at first use.

How was assessed the pancreas consistency as soft, normal, and hard? What was the difference between soft and normal?

- In our study, all procedures were performed by a single, experienced pancreatic surgeon (one of the authors) who performs > 100 pancreatic surgical procedures annually. The consistency of the pancreas parenchyma was assessed by palpation during surgical procedures and information about the parenchyma consistency was derived from the operating protocols. We included this information in the manuscript and in discussion, where we mentioned the subjective nature of the assessment as a weakness of the study. This kind of assessment is also widely used in other available publications. There are efforts to assess pancreatic consistency preoperatively by CT (fibrosis assessment) and elastography, however, they are not routinely used in our department.

Please consider providing data in Table 3 as both numbers (percentages) for better understanding.

- Table 3 has been corrected according to the recommendations.

What does it mean "fluid reservoir"?

- Thank you for the comment. We corrected the expression to “fluid collection”, meaning a limited fluid collection in the peritoneal cavity that does not meet the criteria for an abscess.

Why was it regarded as statistical significance at p values <0.0001, as suggested in Table 3 headline? 

- Statistical significance was P<0.05 but in our study, every significant complication has P<0.0001. To avoid confusion we changed the Table 3 caption, including the p<0.05 as significant.

In Table 3, DGE, bleeding was CR or overall?

- The data refer to patients with CR-PPAP and patients without CR-PPAP, respectively. Thank you for the remark, we corrected clerical errors and entered the correct values.

In lines 122-124, it is said that patients with CR-PPAP have had an increased incidence of severe complications compared with patients with no CR-PPAP. Please consider providing percentages for each group.

-The percentages are now provided.

In line 127, I assume the reference is to Table 3, not Table 4. Please check.

-Thank you for noticing the error, the correct reference is now to Table 3.

Table 4 data should be discussed and explained in the text, and p values should be provided.

-The p values are now provided and the paragraph explaining the data is added.

The data provided in Table 5 are weird. For example, What does "reoperation 61.7% vs. 38.29%" mean? The same issue applies to 90-day mortality and types of procedures. Please check and put the correct values.

- Thank you again for your attentive reading, indeed, by mistake Table 5 data was incorrect. We have now provided the correct values.

Regarding the indications for re-exploration, was it only for collections no bleeding? (as one might assume from lines 133 – 135). Please clarify.

- We now added the data on bleeding management during reoperation in Table 5.

On what basis were selected the factors to be explored as potential predictors for the development of CR-PPA? A few potential important factors such as ASA score, cardiovascular co-morbidities, diabetes, jaundice, cholangitis, preoperative biliary drainage, neuroendocrine vs. adenocarcinoma pathology, type of distal pancreatic stump anastomosis, CRP, PCT, etc, appear to be overmissed. Please explain.

- Thank you for this comment. Our selection of factors for multivariate analysis is based on expert knowledge about the risk factors for the development of CR-PPAP according to ISGPS [literature reference: 5] and other recognized publications [literature references: 4,6,17,25,35], and not on p-value screening. We analyzed factors such as the soft consistency of the pancreatic parenchyma, small (<2 mm) width of the pancreatic duct, histopathological diagnosis of the ampulla of Vater cancer, duration and type of the procedure, intraoperative blood loss, use of neoadjuvant chemotherapy, gender, and BMI (body mass index). Selecting the risk factors (predefined method) was based on this article: https://pubmed.ncbi.nlm.nih.gov/27896874/.

We now added an appropriate explanation in our study – lines 109-115.

It is uncommon that all the statistically significant factors from the univariate analyses were found to be independent predictors. Please comment.

- The risk factors in our study were not chosen based on univariate analysis, but according to the recent recommendations (as in the publication mentioned above), we analyzed the PPAP risk factors well established in the literature and performed the multivariate analysis. We now included an explanation in the manuscript to avoid misunderstanding (lines 109-115 and 169-171).

Figure 4 appears not to illustrate the ampulla of Vater cancer location correctly.

- Figure 4 has been corrected for a hopefully clearer anatomical depiction.

Based on the results of the present study, what is the suggested value for clinical decision-making?

- Currently, there are no generally recognized guidelines on how to prevent CR-PPAP. In our study, octreotide was used perioperatively in selected cases, where the PPAP and POPF risk was considered high due to the pancreatic parenchyma intraoperatively assessed as soft and/or narrow pancreatic duct (<2mm) was found. However, there are conflicting results of studies, with some showing its efficacy in reducing the incidence of postoperative complications, such as POPF, intraabdominal fluid collections, and PPAP, without any difference in mortality rate and others postulating its negative impact on PD results.

Perhaps intensified postoperative supervision and prophylactic use of octreotide in selected patients with risk factors for the development of CR-PPAP mentioned in our study could contribute to the reduction of this complication. We shortly discuss this suggestion.

Another method accepted in selected cases is upfront total pancreatectomy to completely avoid the pancreatic anastomosis creation and PPAP risk. In patients with no other indications for total pancreatectomy present apart from the PPAP/ POPF risk factors (soft tissue, narrow MPD), the decision is made intraoperatively by the surgeon without any general recommendations. In our department, upfront total pancreatectomy is performed at the discretion of the operating surgeon in selected high-risk cases. We now added this to our discussion.

The references should be provided in the format the journal requires

-The references were adjusted to the requirements.

Round 2

Reviewer 1 Report

Comments and Suggestions for Authors

the authors have responded point by point to comments and questions that have significantly improved the quality of the manuscript.

Author Response

Dear Reviewer,

Thank you so much for the positive review. We were happy to improve our work.

Reviewer 2 Report

Comments and Suggestions for Authors

The authors addressed many important issues and concerns raised by the reviewers. However, still to improve:

Major concerns

Although I am not a qualified statistician, it appears strange to me to perform multivariate analyses based only on presumed factors from the literature. To my knowledge, potential predictors should be first explored in the univariate analyses (for the study cohort). After that, the factors with p-values less than 0.05 should be further introduced in the multivariate analyses. A qualified statistician is mandatory to answer this problem. The other reviewer also highlighted these aspects.

Minor concerns

In Table 4, please consider putting the p values column as the last one as in the previous Tables.

In Table 5, please consider providing the p values.  

Author Response

Dear Reviewer,

Thank you for your insightful comments. We corrected the tables according to your comments.

The concern about the statistics is understandable because until now most scientists still apply the univariate analysis to determine the variables with significance, and subsequently perform the multivariate analysis. However, the approach to statistics is now changing. We were happy to see the other reviewer satisfied with our previous response, where we linked an article by Georg Heinze and Daniela Dunkler from Section for Clinical Biometrics, Center for Medical Statistics, Informatics and Intelligent Systems, Medical University of Vienna, Vienna, Austria on this matter. Below are some excerpts:

 “Whatever technique applied, the approach of letting statistics decide which variables should be included in a model is popular among scientists. (…) However, it is not commonly known that there hardly exists any statistical theory which justifies the use of these techniques. (…)Moreover, univariable prefiltering, sometimes also referred to as “bivariable analysis,” does not add stability to the selection process as it is based on stochastic quantities, and can lead to overlooking important adjustment variables needed for control in an etiologic model. Although univariable prefiltering is traceable and easy to do with standard software, one should better completely forget about it as it is neither a prerequisite nor providing any benefits when building multivariable models  (…)Before using variable selection techniques one should critically reflect whether such methods are needed in a particular study at all, and if yes, whether there is enough data available to justify elimination or inclusion of variables in a model just by “letting the data speak.” By contrast, expert background knowledge, for example, formalized by directed acyclic graphs  is usually a much better guide to robust multivariable models.”

We would also like to include a link to the scientific work published previously in an esteemed peer-reviewed medical journal (IF=8.1) by one of the co-authors, where the same statistical method was applied:

https://annalsofintensivecare.springeropen.com/articles/10.1186/s13613-023-01191-0

To the best of our knowledge, as much as this statistical model is not ubiquitous, it is nevertheless correct.

Round 3

Reviewer 2 Report

Comments and Suggestions for Authors

I am not qualified for statistical methods. The study is okay based on the references provided by the authors.